# The Impact of Composites with Silicate-Based Glasses and Gold Nanoparticles on Skin Wound Regeneration

**DOI:** 10.3390/molecules26030620

**Published:** 2021-01-25

**Authors:** Sorin M. Mârza, Klara Magyari, Sidonia Bogdan, Mirela Moldovan, Cosmin Peștean, Andras Nagy, Adrian Florin Gal, Flaviu Tăbăran, Robert Cristian Purdoiu, Emilia Licărete, Sorina Suarasan, Lucian Baia, Ionel Papuc

**Affiliations:** 1Faculty of Veterinary Medicine, University of Agricultural Science and Veterinary Medicine, 400372 Cluj-Napoca, Romania; sorinmarza@yahoo.com (S.M.M.); sidoniabogdan@gmail.com (S.B.); cosmin.pestean@usamvcluj.ro (C.P.); Nagyandras26@gmail.com (A.N.); adrian.gal@usamvcluj.ro (A.F.G.); alexandru.tabaran@usamvcluj.ro (F.T.); robert.purdoiu@usamvcluj.ro (R.C.P.); ionel.papuc@usamvcluj.ro (I.P.); 2Faculty of Physics, Babeș-Bolyai University, 400084 Cluj-Napoca, Romania; 3Interdisciplinary Research Institute on Bio-Nano-Science, Babeș-Bolyai University, 400271 Cluj-Napoca, Romania; emilia_licarete@yahoo.com (E.L.); sorina.suarasan@ubbcluj.ro (S.S.); 4Department of Applied and Environmental Chemistry, University of Szeged, 6720 Szeged, Hungary; 5Faculty of Pharmacy, Iuliu Hațieganu University of Medicine and Pharmacy, 400012 Cluj-Napoca, Romania; mmoldovan@umfcluj.ro

**Keywords:** wound healing, in vivo, Vaseline, immunological assay, histopathology

## Abstract

The silver content of the skin regeneration ointments can influence its regeneration process but in the meantime, it can take the benefit of the antibacterial properties of silver by avoiding the bacterial infection of an open wound. In the current study, the skin healing and regeneration capacity of bioactive glass with spherical gold nanocages (BGAuIND) in the Vaseline ointments were evaluated in vivo comparing the bioactive glass (BG)-Vaseline and bioactive glass with spherical gold (BGAuSP)-Vaseline ointments. Spherical gold nanocages are stabilized with silver and as a consequence the BGAuIND exhibits great antibacterial activity. Histological examination of the cutaneous tissue performed on day 8 indicates a more advanced regeneration process in rats treated with BGAuSP-Vaseline. The histopathological examination also confirms the results obtained after 11 days post-intervention, when the skin is completely regenerated at rats treated with BGAuSP-Vaseline compared with the others groups where the healing was incomplete. This result is also confirmed by the macroscopic images of the evolution of wounds healing. As expected, the silver content influences the wound healing process but after two weeks, for all of the post-interventional trials from the groups of rats, the skin healing was completely.

## 1. Introduction

Skin wound healing is the imperfect reflection of human wound healing processes in animals used as an experimental model. However, they remain essential biological tools in creating of new techniques and approaches in the field [1]. In vivo studies, which had the rat as an experimental model, confirm this model’s positive effects in the wound healing process by increasing the synthesis and the degree of crosslinking of the collagen, expression of the growth factor, proliferation of fibroblasts, formation of blood vessels and contracture of wounds [2,3]. Wound healing occurs in four partially overlapping phases: hemostasis, inflammation, proliferative and remodeling phase [3]. The proliferation phase begins from day 3 alongside angiogenesis and vascularization but at the same time, the probability of infection is high [4]. A viable skin regeneration strategy is to involve tissue engineering that requires the use of high-potential biomaterials to accelerate the angiogenesis process and to ensure better vascularity [3,5]. Biomaterials with gold nanoparticles (AuNPs) can accelerate wound healing through several mechanisms: by decreasing the septic phase of healing by an antioxidant activity [6], through epithelial and mesenchymal cells migration in the injured skin, through myofibroblasts distinction and speeding up the angiogenesis cycle [7]. But the beneficial properties of AuNPs depend on their size, shape and surface. For tissue regeneration, the ideal diameter of AuNPs is between 25 and 100 nm [8,9].

As already mentioned, the probability of infection begins from day 3, creating an important regeneration process problem. Bacterial infection under the dressing extends the inflammation response. Thus the wound dressing must contain antibiotics or antibacterial agents to prevent the bacterial infections [10]. Extending the properties of biomaterials included in regenerative medicine with the antibacterial properties can function sequentially or synergistically in the regeneration cycle. Gold nanoparticles are generally not bactericidal or have a weak antimicrobial effect [11,12,13] but their properties can be improved by adding a bactericide component. A viable strategy to obtain the antibacterial effect is the use of silver nanoparticles [14], which are considered an important antibacterial agent. Mohseni et al. [15] evaluated chronic wound healing using antimicrobial dressings conjugated with silver sulfadiazine and silver nanoparticles. The obtained results indicated that silver nanoparticles are a promising bactericide material in skin regeneration. Silver nanoparticles are used for the synthesis of the spherical gold nanocages (AuIND) [16], allowing to extend the beneficial properties of AuNPs with antibacterial properties.

In a recent study, we investigated bioactive glasses with spherical gold nanoparticles (BGAuSP) embedded in Vaseline ointment, in a concentration of 18 wt%, to accelerate the skin wound healing in an experimental rats model [17]. The obtained results revealed that the BGAuSP-Vaseline ointment accelerates the healing and regeneration of the skin in rats, so it is a promising candidate for wound healing applications. In another recent study of our groups [12], the spherical nanocage (AuIND) was introduced in bioactive glass (BG) matrix and was obtained a great antibacterial activity against *Pseudonomas aeruginosa* and *Staphylococcus aureus* and good proliferation rates of Human keratinocytes cells (HaCaT). Continuing these studies and assuming that the antibacterial effect is preserved during the in vivo environments, we proposed this time to investigate in vivo performance of BG-AuIND-Vaseline ointment by studying wound regeneration rats’ model, by comparing the regeneration ability with the effect of BG-Vaseline and the BGAuSP-Vaseline ointments. It is expected that the silver content of the BGAuIND-Vaseline ointment to influence the regeneration process, since during the in vitro bioactivity tests beside the benefic apatite layer, silver chloride appears.

To ensure a good course of the in vivo trials, in the first step the glasses and glass composites ointments were characterized and the toxicity test were reevaluated by performing in vitro cells viability assessment by using Human kerationocytes (HaCaT). Before starting the in vivo experiment, the microbial load on the skin surface was determined to accurately reproduce the influence of these ointments on the skin healing process and the immunological status of the rats included in the study to determine if the bacterial flora is saprophytic or pathogenic and to eliminate immunosuppressive risk factor specimens. In vivo experiment was performed on Wistar-Lewis rats evaluating the wound healing potential using optical image method followed by histopathology and immunohistochemistry assays.

## 2. Results and Discussions

### 2.1. Glass Composite Ointments Characterization

#### 2.1.1. Glasses Characterization

Given that this work is the continuation of other studies [12,17], the samples quality validation was important. Thus, in the first step UV-Vis absorption spectra of the bioactive glasses were recorded and are illustrated in Figure 1a. The obtained spectra are comparable with the reported ones [12]: the spectrum of BG has an electronic absorption band at 270 and 300 nm assigned to the Si-O-(Si, Ca), P-O-(P-Ca) chemical bonds; the surface plasmon resonance bands of BG-AuSP at 533 nm show the presence of nanoparticle with diameters around 25 nm, while the broadening of the absorption band indicates the polydispersity of AuNPs into the glass matrix structure. On the other hand, the BG-AuIND spectrum exhibits bands at 533, 440 and 290 nm attributed to the presence of spherical gold, silver (I) oxides and the electronic transition of Ag° metallic species, respectively.

To determine the particle size distribution, dynamic light scattering (DLS) was used (Figure 1b). The grain mean size of the obtained material is rather similar, the glass with gold nanoparticles content being slightly larger.

In a previous study [12], human keratinocytes (HaCaT) cells, which are the main cells type in the epidermis, were used for testing the in vitro cells viability. Comparable results were obtained by repeating the cell viability assay. It can be seen that the gold nanoparticle content stimulates cell proliferation (Figure 2) and due to silver nanoparticle content, the cells proliferation rate is better in the case of BG-AuIND.

#### 2.1.2. Ointments Characterization

In the applicable ointment the active ingredient should be distributed uniformly. Therefore, the homogeneity of the distribution of the dispersed microparticles in the vehicles was evaluated. The averages of particle number in the three samples were 15.1 for BG-Vaseline, 15.9 for BGAuSP-Vaseline and 15.6 for BG-AuIND-Vaseline, with an RSD (relative standard deviation) of 3.75%, 4.54% and 4.48% respectively. Thus, it can be considered that the microparticles are uniformly distributed in the vehicle as, in all cases, RSD was below 5% (Figure 3) [18].

The X-ray diffraction (XRD) patterns of the investigated glass samples show the presence of an amorphous structure, the main diffraction peaks, which are well-defined correspond to the AuNPs in the glass structure [12,19]. The Vaseline presents the typical XRD pattern of semi-crystalline petrolatum structure with diffraction peaks at 2θ = 21.8°, 23.5° and with a broad signal at 18.3° (Figure 4a). After introduction glass into the Vaseline, in the XRD pattern of ointment the broad signal is shifting to a higher value from 18.3° to 19.7° and the diffraction peak at 23.5° is broadening, these spectral changes suggesting the existence of a bond between glass and Vaseline. The obtained FT-IR spectrum recorded for Vaseline is typical for petrolatum, in which can be identified the following signals: C=O bending (721 cm^−1^), CH_3_ bending (1375 cm^−1^), C-H bending (1471 cm^−1^) and C-H stretching (2848 and 2919 cm^−1^) (Figure 4b) [20]. In the spectra of the glass-Vaseline ointment beside the characteristic bands of Vaseline, the typical signal of the silicate network appears at 1080 cm^−1^ [21].

### 2.2. Immunological and Bacteriological Assays

Given that the immune status can affect the skin healing quality and regeneration process and speed, in vitro blast transformation assay was used (Table 1). The most intense decrease in the blast transformation index was found in the group treated with BGAuSP-Vaseline ointment. The results indicated a 22.58% decrease in the stimulation index (SI), from 96.62% in the untreated control group to 74.8% in the BGAuSP-Vaseline. In the lots treated with BGAuNPs-Vaseline ointment, ConA induced the lowest stimulation rates, decreasing by 26.94% compared with the control lot. The SI for blast transformation recorded in the groups treated with BG-Vaseline ointment were similar to those treated with BGAuSP-Vaseline ointment and BGAuIND-Vaseline ointment in vivo, with values ranging from 80.50 to 82.00%.

To sum up, these results demonstrate the increased efficacy of the ointments used in rat healing and regeneration process. In skin pathology, these ointments can be considered as potential therapeutic formulas. None of the products tested showed an immune rejection response.

A bacteriological examination was also performed to see if there is a bacterial load on the skin level, before and after the skin defects, that could influence the healing and regeneration process (Figure 5). After incubation, on macroscopic examination, it was observed that round, smooth, small and medium-sized colonies, non-hemolytic pigmented in white Cretaceous, developed on the culture medium. From the grown colonies, smears were performed which were stained Gram and then microscopically examined. In the microscopic examination, bacterial germs belonging to the *Staphylococcus intermedius* genus, a saprophytic germ on the skin, were identified. The difference between the colonies raised from samples collected before surgery and those collected 7 days after surgery is given only by their size and not by the presence of other germs, those on the first day being smaller compared to those on day 7- of post-interventional.

In conclusion, we can say that a pathogenic bacterial microflora that could have influenced the emptying and regeneration process was not identified at the skin level. The identified bacterial flora was normal, with the role of protective barrier.

### 2.3. Skin Regeneration Assessment

The healing potential was evaluated using the optical images of the wound on different days after surgical intervention (Figure 6). Thus, macroscopic analyses of the healing process of skin defects in rats showed that the skin defect’s closure began on day three post-intervention, being visible the healing crust in all 3 groups. On day 11 post-intervention, the skin defect closure process is more advanced for the groups treated with BG-Vaseline and BGAuSP-Vaseline than the BGAuIND group where the defect is still visible. After 15 days of post-interventional in all three groups, the skin healing process is completed and the place where the defect was created is still visible. To have a visual aid regarding the healing process, for each picture analyzed, a surface plot was performed (Figure 7).

Calculating the percent of wound closure for the defect treated with BG-Vaseline, BGAuSP-Vaseline and BG-AuIND-Vaseline ointments can be seen that the wound defect treated with BG/BGAuSP content regenerates faster in the first week of treatment than the one treated with BGAuIND-Vaseline (Figure 8). Worth be mentioned that in all three cases the wound was closed after two weeks. This suggest that silver content in the BGAuIND [12,16] influences the speed of the regeneration process. It is known that AgNPs have strong antibacterial effect excessive agglomeration of these nanoparticles sometimes leads to cell death [22]. In a previous study, we demonstrated the antibacterial effect of the BGAuIND against multi-drug resistant *Pseudonomas aeruginosa* and *Staphyloccoccus aureus* [12]. Considering that the burn patients have an increased risk of developing infections and the *Pseudomonas aeruginosa* is a type of germ present in hospitals [23], the viable strategy would be to use the ointments with BGAuIND to treat these patients. Thus, a study of this possible application is the subject of a future study.

### 2.4. Histopathological Analysis of Skin Defects during the Wound Healing Process (Day 8)

The materials used was evaluated during the healing process on the day 8 by histological analysis compared with each other, providing information regarding the regeneration process during the regeneration. In the following subsection we present in detail the histopathology results on each separate lot.

#### 2.4.1. Histological Analysis on Day 8 of Lot 1 Treated with BG-Vaseline

Histological examination of the rat sections in the first batch treated with BG-Vaseline ointment revealed the following morphological aspects: there is a regenerative reaction in the area of the biomaterial application dominated by intensely vascularized granulation tissue, with numerous new-formed vessels, the fibroplasia reaction is accompanied by a discrete cellular infiltrate consisting of rare young neutrophils, macrophages, lymphocytes and hemorrhages (Figure 9A). In the dermis’ granulation tissue, an acidophilic-amorphous material is identified multifocal, possibly the intradermal applied biomaterial (Figure 9B). The surface epithelium covers the entire defect created along its entire length, adhering to almost the non-forming granulation tissue; there are subepidermal spaces only in patches. The regenerated epidermis presents with keratin on the surface, the crust showing only in patches on the surface of the epidermal keratin.

#### 2.4.2. Histological Analysis on Day 8 of Lot 2 Treated with BGAuSP-Vaseline

Histological examination of the rat sections in the second batch treated with BGAuSP-Vaseline ointment revealed the following morphological aspects: In the area where the defect was created, a complete re-epithelialization is noticed. In this region, an acanthosis epidemic is observed, compared to the epidermis in the area bordering the defect (Figure 9C). A well-contoured connective tissue is identified in the dermis, following the regeneration, with the presence of hair follicles and their annexes (sebaceous glands). The regenerated dermis does not present the specific aspects of granulation tissue, respectively numerous neoformation vessels and fibroblasts. The formed connective tissue has a differentiated, mature appearance, without inflammatory reaction (Figure 9D). At the surface of the epidermis, the crust can be identified.

#### 2.4.3. Histological Analysis on Day 8 of Lot 3 Treated with BGAuIND-Vaseline

Histological examination of the rat sections in the third batch treated with BGAuIND-Vaseline ointment revealed the following morphological aspects: At created defect, there is bleeding that practically fills the created cavity. At the surface of the area with hemorrhage, re-epithelialization is observed along the entire length of the defect (Figure 9E). A discrete fibroplasia reaction is observed in the defect’s marginal areas, respectively granulation tissue with a discrete infiltrate dominated by rare neutrophils, macrophages, especially siderocytes and lymphocytes (Figure 9F).

Comparing the results obtained from histopathological examination of the skin on day 8 postoperatively shows that the wound of the group treated with BGAuSP-Vaseline heals faster than the BG-Vaseline and BGAuIND-Vaseline-treated groups.

### 2.5. Histopathology and Immunohistochemical Analysis

After the in vivo experiment, the healing potential of used materials was evaluated by histological and immunohistochemical analysis compared with the simple Vaseline as a positive control.

#### 2.5.1. Histological Analysis of Lot 1 Treated with BG-Vaseline

Histological examination of the control sections of the first rat in the first lot revealed the following morphological aspects: The surgical wound does not appear completely in the histological section but it is observed in the present part that the deep layers of the dermis and the muscle of *Panniculus carnosus* are replaced by an abundant granular tissue, partially oriented and infiltrated by a few histiocytes (Figure 10A,D). The immunohistochemical analysis shows the granulation tissue in which multiple blood vessels and spindle cells are visible, identified as myofibroblasts, which are positive for α-SMA immunostaining (Figure 10G). Histological analysis of the control sections performed on samples taken from the second rat from the first lot confirms the existence of the same structural elements as for the first rat, claiming that histiocytes infiltrated in the granulation tissue were centered around contaminated hair rods.

Within the histological images obtained from the sections of the first rat’s defects treated with BG-Vaseline ointment, it is found that the surgical wound has been healed completely, the epidermis is normal, becoming completely structured; the superficial dermis is infiltrated with rare lymphocytes and histiocytes; the collagen has a morphology similar to that of additional normal tissue in terms of the thickness of the beam, dyeing and tissue orientation; hypodermis does not show an inflammatory reaction and pilosebaceous units are normally regenerated in the induced defect; *Panniculus carnosus* muscle is interrupted and replaced by granulation tissue in the regeneration mixture of muscle fibers (Figure 10B,C,). Immunohistochemical analysis showed the presence of granulation tissue, in which multiple blood vessels and spindle cells, known as myofibroblasts, were found to be positive for α-SMA immunostain.

Histopathological analysis of the second rat of this lot, treated with BG-Vaseline ointment, shows the presence of the same morphological elements with completely healed surgical wound, regular epidermis but at the level of the superficial dermis, focally, near to the excision area, a hypercellular layer is observed made from lots of fibroblasts, lymphocytes, histiocytes and mast cells, the fibroblasts being guided parallel with the epidermis and separated from the rare capillaries; collagen has a morphology similar to the one found in the first rat and the deep dermis or hypodermis has focal infiltration with rare lymphocytes, plasma cells and mast cells with partially regenerated pilosebaceous units; the *Panniculus carnosus* muscle near the defect is interrupted and replaced with scar tissue infiltrated with leukocytes similar to those in the deep dermis (Figure 10E,F).

Immunohistochemical examination revealed the presence of scar tissue in the muscle of *Panniculus carnosus* containing many blood vessels and spindle cells, identified as myofibroblasts, positive for immuno-staining with α-SMA (Figure 10H,I).

In conclusion, in the group treated with BG-Vaseline ointment, the wound is completely healed. The epidermis is normal. The dermis has focal infiltrates with rare lymphocytes, plasma cells and mast cells. The pilosebaceous units are partially regenerated. In the control group, however the surgical wound is not completely healed. The dermis and muscle panniculus carnosus is replaced by abundant granular tissue, partially oriented.

#### 2.5.2. Histological Analysis of Lot 2 Treated with BGAuSP-Vaseline

Histological examination of the defective sections considered in rat 1 of the 2nd lot found the following aspects: the surgical wound is partially healed, the epidermis has focal areas of discontinuity and is covered by a sero-cellular crust and around the area of defective, the epidermis is unilayered, being regenerated. The superficial dermis is focally infiltrated with numerous neutrophils, especially near the epidermal defect and the adjacent areas are replaced by abundant scar tissue. The pilosebaceous units in the hypodermis are not present in the defect area. The deep dermis and *Panniculus carnosus* muscle are infiltrated with numerous neutrophils mixed with macrophages, focally forming a pyogranulomatous reaction centered on an amorphous necrotic material, lymphocytes and mast cells (Figure 11B,D). Immunohistochemical examination of scar tissue contains few positive cells for α-SMA immuno-staining.

At the histological examination of the control sections performed on the samples taken from the 2nd rat from the 2nd group, the surgical wound was fully healed, the epidermis was regular and the pilosebaceous units were partly regenerated. The superficial, deep dermis and focal *Panniculus carnosus* muscle are replaced near the dermal defect area with granulation tissue, sometimes in excess, partially oriented, low in collagen fibers, containing focal discrete granulomas, centered around hair shafts. Immunohistochemical analysis shows granulation tissue in which the presence of a cell population is observed, most cells, except the leukocyte population, which consists of spindle-shaped cells, represented by myofibrils, positive for α SMA immuno-staining (Figure 11G).

In the histological images of the sections of the defects treated with BGAuSP-Vaseline ointment from the first rat, it is observed that the surgical wound was completely healed, the epidermis is regular, partially present in the section, the superficial dermis, the deep one and the *Panniculus carnosus* muscle are focally replaced near the area of the defect with partially oriented granulation tissue (excess), low in collagen fibers, that contains multifocal granulomas of inert foreign body, centered on a foreign-granular, basophilic material, interpreted as a part of the tested product. The numerous histiocytes found at the level of the injury present intracytoplasmic granular, heterogeneous, black material, also being considered as part of the tested product. Focally, especially in the deep dermis, numerous eosinophils are present mixed with rare mast cells. The pilosebaceous units are normally regenerated in the excision area (Figure 11B,C).

In the immunohistochemical examination most of the cells mentioned above, except for the leukocyte population, in the granulation tissue that replaces the dermis and skin muscle, are present fusiform cells, identify as myofibril, positive α SMA immuno-staining.

Histological images made from the defective sections treated with BGAuSP-Vaseline ointment from the 2nd rat show the completely healed appearance of the surgical wound, the regular epidermis, partly present in the section, the superficial, focal dermis, near the defect area presents numerous fibroblasts, oriented parallel to the epidermis, separated by rare capillaries, lymphocytes and mast cells. The collagen present has a morphology similar to the additional normal tissue in terms of the thickness of the beams, dyeing and tissue orientation. The deep, focal dermis is infiltrated with rare lymphocytes, plasma cells and mast cells. The pilosebaceous units are partially regenerated and the skin muscle, next to the defect, is interrupted and replaced with scar tissue infiltrated with rare mast cells (Figure 11E,F).

In the immunohistochemical examination, most of the fibroblasts described above are positive for α SMA immuno-staining (Figure 11H,I).

In conclusion, in the group treated with BGAuSP-Vaseline ointment, the wound is completely healed. The epidermis has a normal structure. The dermis has numerous fibroblasts oriented parallel to the dermis and the pilosebaceous units are completely regenerated. While the wound is partially healed in the control group, the epidermis is focally discontinuous, the dermis is replaced by abundant scar tissue and the pilosebaceous units are partially regenerated.

#### 2.5.3. Histological Analysis of Lot 3 Treated with BGAuIND-Vaseline

Histological examination of the sections of the control defects from the 1st rat in the 3rd group showed the following aspects: the surgical wound is completely healed, the normal epidermis, the missing pilosebaceous units, the superficial dermis, the deep dermis and the *Panniculus carnosus* muscle are replaced near the area. Defect with granulation tissue, in excess, partially oriented, poor in oriented collagen fibers (Figure 12A).

In the immunohistochemical examination in the excess granulation tissue described above, most spindle-shaped cells are myofibroblasts, positive for α SMA immuno-staining (Figure 12G).

In the histological examination of the control sections performed on the samples taken from the 2nd rat from the 3rd group, the surgical wound is partly healed, the epidermis, focally appears discontinuous, covered by a sero-cellular crust; and in other areas, it is composed of a single cell line, in the process of regeneration. The superficial and deep dermis are replaced near the defective area with excess granulation tissue, partly oriented, poor in oriented collagen fibers, infiltrated by numerous lymphocytes. Pilosebaceous units are absent in the area where the excision was made. The skin muscle is not present in the section.

Immunohistochemical examination reveals the same morphological structures described in the sections taken from the control samples from the 1st rat, that being the excess granulation tissue with myofibroblasts, positive for α-SMA immuno-staining (Figure 12G).

In the histological images of the sections from the defects treated with BGAuIND-Vaseline ointment from the 1st rat, it is observed that the surgical wound was completely healed, the epidermis is regular, moderately hyperplasia at the area where the excision was made, the pilosebaceous units partly regenerated in the defect induced, the superficial and deep dermis were replaced by excess scar tissue, rich in collagen fibers oriented parallel to the epidermis, tissue infiltrated by a moderate number of lymphocytes, plasma cells and histiocytes. The skin muscle near the defect is interrupted and replaced with granulation tissue similar to that described from the dermis (Figure 12B,C).

In the scar tissue’s immunohistochemical examination, few cells are observed that are positive for α SMA immuno-staining (Figure 12H,I).

Histological examination of the sections made from the defects treated with BGAuIND-Vaseline ointment from the 2nd rat shows the completely cured appearance of the surgical wound, the normal epidermis and the absence of pilosebaceous units. The superficial and deep dermis were replaced in the defective area with excess granulation tissue, partially oriented, poor in oriented collagen fibers. The skin muscle is not present in the section (Figure 12E,F).

Immunohistochemical examination reveals the presence of morphological elements faced in the 1st rat treated with BGAuIND-Vaseline ointment.

To sum up, based on the analyzed histopathological images it can be affirmed that in the witnesses from the 2nd group the skin defects are partially healed, the epidermis has focal areas of discontinuity and is covered by a sero-cellular crust and around the defect area, the epidermis is unistratified, being in the process of regeneration. The superficial dermis is focally infiltrated with numerous neutrophils, especially near the epidermal defect and the adjacent areas are replaced by abundant scar tissue. The pilosebaceous units in the hypodermis are not present in the defect area.

At the level of defects treated with BGAuSP-Vaseline ointment, the surgical wound has been completely healed, the epidermis is regular but the dermis, hypodermis and *Panniculus carnosus* muscle are focally replaced near the defect area with excess granulation tissue, partly oriented, poor in collagen fibers, containing multifocal granules of inert foreign body, centered on a foreign-granular, basophilic material, interpreted as part of the tested product.

At the level of control defects of the 3rd lot, the surgical wound was fully healed, the epidermis was normal but the pilosebaceous units were absent and most of the dermis and hypodermis were replaced in the area of the defect with excess granulation tissue, partially oriented, low in oriented collagen fibers.

At the level of defects treated with BGAuIND-Vaseline ointment, the surgical wound was completely healed, the epidermis is moderately hyperplastic at the area where the defect occurred, the partially regenerated pilosebaceous units, the superficial and deep dermis were replaced by excess scar tissue, rich in collagen fibers oriented parallel to the epidermis, tissue infiltrated by a moderate number of lymphocytes, plasma cells and histiocytes. The skin muscle, *Panniculus carnosus* near the defect is interrupted and replaced with granular tissue similar to that described in the dermis.

In conclusion, in the group treated with BGAuIND-Vaseline ointment, the epithelium is moderately hyperplasia. The pilosebaceous units are partially regenerated and the collagen fibers are oriented parallel to the epidermis. While in the control group, the pilosebaceous units are absent. The dermis and the panniculus carnosus muscle are replaced with excess granulation tissue, partially oriented.

## 3. Materials and Methods

### 3.1. Materials

The precursors used for the glass synthesis were tetraethyl orthosilicate (TEOS, ≥99%, Merck, Hill Road Kenilworth, NJ, USA), triethyl phosphate (TEP, ≥99%, Merck) and calcium nitrate tetrahydrate (Ca(NO_3_)_2_·4H_2_O, ≥99%, Lach-Ner, Neratovice, Czechia), hydrolyzed in presence of nitric acid (HNO_3_, 65%). Hydrogen tetrachloroaurate (III) hydrate (HAuCl_4_·3H_2_O, 99.99%, Sigma-Aldrich, St. Louis, MO, USA), trisodium citrate dihydrate (ACS, 99.0%), cetyltrimethyl ammonium bromide (CTAB, Sigma-Aldrich), sodium borohydride (NaBH_4_, purum ≥ 96%, Sigma-Aldrich), silver nitrate (AgNO_3_, Penta, Frankfurt, Germany), sodium chloride (NaCl, Poch basic, Gliwice, Poland), Pluronic F127 (powder, BioReagent, suitable for cell, Sigma-Aldrich) were used for AuNPs synthesis. Ultrapure water and absolute ethanol were used throughout the whole experimental process. All chemicals were used as received without further purification. For the ointment Vaseline (Merck, Darmstadt, Germany) was used.

For the surgery, the following materials were used: biopsy-punch with a diameter of 5 mm (by relaxing the skin, the excision has a diameter of 6 mm); antiseptic solutions: betadine and sanitary alcohol 70%; scalpel with fixed blade; surgical and hemostatic forceps; scissors; non-absorbable needles and suture threads (Nyllion 4.0); sterile swabs for microbiological sampling, heparin tubes, sterile glass tubes, trimmer; depilatory cream; sterile bandage; elastic bandage and superglue.

### 3.2. Glass Composite-Ointments Preparation and Characterization

#### 3.2.1. The Synthesis and Characterization of Glass Composites with AuNPs

In the sol-gel prepared bioactive glasses matrix (BG, 60SiO_2_·32CaO·8P_2_O_5_ mol%) was introduced different shaped AuNPs (60SiO_2_·31.85CaO·8P_2_O_5_·0.15Au_2_O, the gold amount is conventionally indicated in the oxidic form Au_2_O), namely spherical gold nanoparticles (BGAuSP) and spherical gold nanocages (BGAuIND) as described elsewhere [12,19]. For the BG samples, the reactants were added consecutively after 1-h intervals, under continuous stirring. In the final step, the colloidal gold solution was added and stirred for 1 h.

The spherical nanoparticles (AuSP) were obtained using the Turkevich-Frens method [24]. HAuCl_4_∙3H_2_O solution was prepared (10^−3^ M). After this, the solution was heated until the boiling point of water. Afterward, 38.8 × 10^−3^ M trisodium citrate solution was instantly added to the boiling gold precursor solution using a 10:1 volume ratio. The whole procedure continued for 30 min and then it was cooled to room temperature. The spherical gold nanocages (AuIND) were prepared by adapting a well-known method in the literature [16]. The Ag nanoparticles were prepared via chemical reduction method, where NaBH_4_ was used as a reducing agent and trisodium citrate as a stabilizing agent. The next step was the galvanic replacement reaction between the as-obtained Ag nanoparticles and HAuCl_4_·3H_2_O solution. NaCl was used to eliminate the remaining Ag^+^ in form of AgCl.

For further application, AuNPs were stabilized using a Pluronic F127 block copolymer solution of 0.5 × 10^−3^ M. The solution containing the AuNPs was stirred for 20 min, followed by an ageing process of 24 h to assure the adherence of the polymer molecules. The excess of Pluronic F127 was removed by collecting the supernatant liquid from the centrifugation tubes, which were held for 30 min at 12,000 rpm. The authors would like to mention also that all the stabilized AuNPs were used during the preparation process of the bioactive network. Consequently, the real Au content can be considered the initially use done.

The gelation was achieved in ≈2 days at 37 °C and the gels were aged 3 days at 37 °C. The maturated gels were dried at 110 °C for 24 h and thermally treated at 500 °C for 2 h. All analyses were performed on powder samples, which were milled by hand using Agate mortar. The glass samples’ granulation was similar.

To verify the optical response of the colloidal AuNPs (data not shown) and AuNPs in the glass matrix, *UV-vis spectroscopy* measurements were used. Absorption measurements were performed with an Analytic Jena Specord 250 plus UV-Vis spectrometer with a spectral resolution of 2 nm.

To determine the particle size distribution of glasses *dynamic light scattering* (DLS, Malvern Instrument Zetaiser Nano-ZS) measurements were achieved at 25 °C temperature. The results were averaged from three measurements.

#### 3.2.2. Glass Composite Ointments Formation

The glass composite ointments (BG-Vaseline, BGAuSP-Vaseline and BGAuIND-Vaseline) were obtained by dispersing powders in 18% (*w*/*w*) concentration into neutral liquefied medical Vaseline (45 ± 2 °C) representing the experimental product used in this research. The vehicle was added in geometric progression, with 1 min of homogenization after each step and continuous stirring until the ointment reached room temperature.

The number of microparticles per ten microscopic view fields was counted to appreciate the homogeneity of the distribution of the dispersed microparticles in the vehicle. Three different samples, taken from three different parts of the ointment were examined at 40× magnification using an optical microscope (Olympus, Tokyo, Japan, CX22LED). To obtain the same thickness of product samples, the same quantity of product was placed onto the microscopic slide and pressed with a cover-glass down until the ointment was spread to a circular diameter of 1.5 cm. According to the Pharmacopeial Forum product quality-test recommendations, the homogeneity test’s acceptance criteria can be set at a maximum RSD (relative standard deviation) of 5% from the average number of microparticles in each sample.

For the structural characterization of ointments, *X-Ray diffraction pattern* (XRD) and *Fourier Transform Infrared* (FT-IR) Spectroscopy measurements were performed. The XRD patterns were recorded with the Shimadzu XRD 6000 diffractometer (Kyoto, Japan) using CuKα radiation (λ = 1.54 Å) with Ni filter, in 2θ range from 10° to 80° with a speed 2°/min. The FT-IR absorption spectra were recorded in reflection configuration with a Jasco 6000 (Jasco, Tokyo, Japan) spectrometer, at room temperature, in the range 400–4000 cm^−1^; spectral resolution of 4 cm^−1^; using the well-known KBr pellet technique.

#### 3.2.3. In Vitro Cell Viability of Glass Composites with AuNPs

Cell viability assessment was carried out on Human keratinocytes cells (HaCaT, Cell Line Service, Eppelheim, Germany) similar to our previous experiments [19,25]. The glass samples’ cytotoxic effect was assayed using WST-1 dye (water soluble tetrazolim salt, Millipore, Burlington, MA, USA), with a method based on the enzymatic cleavage of the tetrazolium salt WST-I to formazan by mitochondrial dehydrogenases active in the living cells. The HaCaT cells density was 10^4^ cells/well. Viability after 24 h was determined by measuring the absorbance at 440 nm, using a microplate reader (Flostar Omega, BMG Labtech, Offenburg, Germany). All samples in three concentrations were tested in triplicate and for the positive control were used cells without samples.

### 3.3. Animal Care and Use

A total of 30 rats, adult females, weighing about 250 g, from Muridae family, the laboratory rat species, the Wistar-Lewis line [26], represented the biological material used to carry out the experiment. This line was chosen because it exhibits an increased degree of docility in experimental work, has a reduced sensitivity to bacterial infections, to the appearance of spontaneous tumors and has a good capacity ability to adapt to captive breeding [27]. The animals were purchased from the Experimental Medicine Center within the University of Medicine and Pharmacy Iuliu Haţieganu, Cluj-Napoca, Romania and then transferred and housed in the Establishment for breeding and use of laboratory animals of the Faculty of Veterinary Medicine, Cluj-Napoca, Romania where the experiment took place. The rats studied benefited from standard maintenance and feeding conditions, a temperature of 23 °C, humidity cycles of 55% and light/dark cycles of 12 h, according to ISO 10993-2 [28]. They were fed with standard granulated food for rodents and had unlimited access to water. Out of the total number of 30 rats, 3 groups of 10 individuals were formed. The experiment was approved by the Bioethics Committee of the University of Agricultural Sciences and Veterinary Medicine Cluj-Napoca no. 113/18.06.2018 and authorized by the Sanitary-Veterinary and Food Safety Department, Cluj-Napoca through the Project Authorization no. 129/05.07.2018.

### 3.4. Surgical Procedure

Each rat was weighed and anesthetized with a mixture of Ketamine and Xylazine (Xylazin Bio 2%, Bioveta, Ivanovice na Hané, Czech Republic, 6 mg/kg and Ketamine Narkamon Bio, Bioveta, Ivanovice na Hané, Czech 60 mg/kg) [29], injected intraperitoneally, the dose being calculated for each rat. After anesthesia, from each rat was collected blood from the infraorbital sinus for analysis to see the state of their immunity. After that, the place of the excision was prepared for the surgery by trimming the hair in the dorsal area of the thorax from T1 to L1. Special attention was granted to the skin in the wither’s region between the two shoulders up to the basin (Figure 13A). After trimming, a hair removal cream was applied and allowed to act for 3 min to ensure complete removal of the hair. The requirements of applying the depilatory cream to stay in contact with the skin for 5 min have not been respected to avoid burns. The removal of the depilatory cream was done with a special plastic scraper. The antisepsis of the area was obtained by cleaning it with sterile pads soaked with betadine solution, after that, sterile pads soaked in 70% sanitary alcohol were used to obtain a better effect and to remove the color induced by the tincture of iodine.

Three lots of rats were studied, which were performed 2 dermal excisions in the dorsal region of the withers, on one side and the other side of the spine (Figure 13). To prevent the skin contraction, due to the strong action of the panniculus carnosus muscle, which is highly developed in this species, semi-hard silicone threads with a thickness of 1 mm were made which were fixed to the skin around the excision. The silicate rings were obtained by cutting some spheres with a diameter of 15 mm using a metal drill press, which in turn were cut inside with a metallic drill press of 6 mm, thus forming a silicate ring with a total diameter of 15 mm and the diameter of the hole of 6 mm. These silicate rings in the surgical operation were done only after they were sterilized in a disinfectant solution. Afterward, they were packed in sterile envelopes.

After the preparation of the area, dermal excisions were performed in the following steps: the rat was fixed in the lateral position, then the skin in the dorsal region, on the line of the spine was trapped and pulled with the help of two hemostatic forceps, obtaining a skin fold (Figure 13B). The dermal excisions were performed using a biopsy-punch, fixed at a distance of approximately 8 mm from the edge of the skin fold (Figure 13C). The 2 dermal excisions were obtained by slow rotational movements of the biopsy-punch that crossed both layers of the skin fold. The distance between the 2 excisions is about 16 mm, as a result of the detachment of the skin fold [29]. After the dermal excisions were performed, they were cleaned with a sterile pad. To minimize the constricting effect of the panniculus carnosus muscle on each excision a silicate ring was applied. On the side of the silicate ring that comes in contact with the skin, it was applied superglue for immediate fixation, which subsequently allowed them to be sutured with non-absorbable threads, so that each silicate ring was fixed to the skin with 4 symmetrical suture threads (Figure 13E) [29].

The ointment was applied according to the following working protocol—each rat was fixed in the sterno-abdominal decubitus, its hind limbs coming from the right side of the researcher, so that the dorsal region with the 2 excisions to remain free, offering direct access for the application of the ointment. The dermal excision from the left side is the witness excision for all three lots. On this was applied simple Vaseline. The dermal excision on the right side is the excision on which the experimental product was applied, specifically the ointment with different BG compositions.

Therefore, BG-Vaseline was applied to the first group, BGAuSP-Vaseline was used for the second group and ointments with BGAuIND-Vaseline were used for the third group. After applying the simple Vaseline and the experimental product, each rat was bandaged with a sterile elastic bandage to prevent infection of the wounds (Figure 13F). On days 4 and 7 of the experiment, a new application of ointment was used, following the same protocol and at the same concentrations, after that, the rats were bandaged. In order to photograph the excisions (days 1, 3, 6, 11 and 15), the application of the ointment (days 4 and 7) and the dressing were done using inhalation anesthesia (Isolflurane).

Postoperative, Tramadol was administered, subcutaneously, in a dose of 12.5 mg/bw. Tramadol analgesia was accomplished during the first 3 days, once a day. Afterward, pain monitorization was done during the whole experimental study period and analgesia administered when needed.

On the 8th postoperative day, skin samples were taken from the excisions of one rat from each batch. After the wounds were completely healed at 15 days from the surgery, each rat was euthanized by an overdose of anesthetic and cervical dislocation. From the place of each dermal excision, samples of skin tissue from the interesting spot were taken, where the scar formed and normal tissue around it, about 1 cm for the histopathological examination.

### 3.5. Immunological Assays

Considering that the immune status can influence the quality and speed of wound healing, on the day of the intervention but also on the 7th day after the intervention, it was collected from each rat approximately 2 mL of blood from the infraorbital sinus, to establish their health status through assessment of the immunity. The adaptive cell-mediated immune response was measured in the in vitro blast transformation assay by classical mitogens: phytohemagglutinin obtained from *Phaseolus vulgaris* (PHA M), concanavalin produced by *Canavalia ensiforis* (ConA) and lipopolysaccharide from the wall of Gram-negative bacteria (LPS) at the untreated witness group compared with the groups treated with BG-Vaseline, BGAuSP-Vaseline and BGAuIND-Vaseline. In vitro, the immunological status design used four variants: control, treated with PHA M, treated with ConA and treated with LPS, which were tested in whole blood cultures in the proliferative response of lymphocytes to mitogens (RPMI). The cultures were incubated 48 h at 37.5 °C and 5% CO_2_. Growth was estimated by the calculation of stimulation indices (SI%) based on the glucose consumption test. Glucose concentrations were measured in the initial medium and in the end of the incubation period, using a standard (100·mg·dL^−1^) glucose solution, using an auto-toluidine colorimetric test and read in a spectrophotometer at 610·nm wavelength (SumalPE2, Karl Zeiss, Oberkochen, Germany), using the reagent as a blank.

### 3.6. Bacteriological Test

Microbiological samples were collected from the skin level, both on the day of the intervention and on the 7th day after the intervention by buffering the wounds. This was done in conformity with the following steps: the collection of biological samples, the shipment, their macroscopic examination, microscopic examination, sowing on culture medium and identification of germs based on certain characteristics. Samples were collected after the hair in the area of choice of excision was cut. Therefore, the sample was made from the 3 groups of rats using sterile swabs: the control group treated with BG-Vaseline ointment, the group treated with BGAuSP-Vaseline ointment and the group treated with BGAuIND-Vaseline ointment. After collection, the samples were seeded in Petri dishes on blood agar and incubated for 24 h at 37 °C. The gram-colored smears were performed on the medium grown clones, which were examined microscopically (Optika B-350).

### 3.7. Measurements of Wound Size Reduction

To assess the wound healing process, pictures were taken at 1, 3, 6, 11 and 15 days. The pictures were captured using an iPhone 8 with a 12MP camera without flash. For wound healing evaluation we have used a free software (ImageJ 1.53). The wound area was delimitated using the freehand drawing tool, the scale was set at 10 unit/mm [30]. The surface plot was performed by converting the images to grayscale, showing a 3D histogram of the wound’s pixel distribution. The amount of wound closure was calculated using the initial wound defect area (*A*_0_) and the wound defect area at each time point (*A_t_*) according to the following formula:$$Wound\;closure\;\left(\%\right)=\frac{A_{0}-A_{t}}{A_{0}}\times 100.$$

### 3.8. Histological and Immunohistological Methods

At the end of the experimental period, the animals were sacrificed and skin samples were harvested for histological examination. Skin samples were fixed in 10% buffered neutral formalin, embedded in paraffin, sections were made at 4 micrometers and the slides were stained by Haematoxiline–Eosine (HE) method. The slides were examined under a microscope Olympus BX 51. The images were taken with Olympus UC 30 digital camera and processed by a special image acquisition and processing program: Olympus Stream Basic. The angiogenesis was assessed by α-SMA immunostaining using a mouse monoclonal antiα-SMA antibody (clone 1A4) (Abcam ab76549) in a 1:800 dilution. The reaction was carried by a Leica Bond-Max (Leica Microsystems, Wetzlar, Germany) automated immunostainer using a polymer-based detection system (Leica Biosystems, Wetzlar, Germany) with 3,3′-Diaminobenzidinew (DAB) as the chromogen. Masson-Goldner’s trichrome (TM)-stained method was used for histological examination during the wound healing process (day 8).

### 3.9. Statistical Analysis

All data reported in cell viability assay and wound regeneration are as the mean ± SD. The triplicate (*n* = 3) values obtained for cell viability were analyzed by two-way analysis of variance ANOVA. Statistical significance was at *p* < 0.05 in all cases. Statistical values were obtained using GraphPad Prism 8.0 software. For wound regeneration, the t-student test was performed. The t-student test with two tail shows statistical significance when comparing the BG-Vaseline and BGAuIND-Vaseline group, *p* < 0.05 and t-stat > t-crit. There is no significant difference between the BG-Vaseline and the BGAuSP-Vaseline group (*p* = 0.05) and between the BGAuIND-Vaseline group and the BGAuSP-Vaseline group (*p* > 0.05). The null hypothesis is rejected only in the case of BG-Vaseline and BGAuIND-Vaseline group comparison

## 4. Conclusions

The BGAuIND with antibacterial effect was introduced into Vaseline ointments and was investigated in vivo for the future skin regeneration trials comparing with the BG-Vaseline and BGAuSP-Vaseline ointments regeneration capacity. The reproducibility of the obtained BG, BGAuSP and BGAuIND structures has been proven, the glass samples presenting an amorphous structure with specific FT-IR vibrational bands. The UV-Vis absorption spectra of BG-AuIND showed the presence of spherical gold, silver (I) oxides and the electronic transition of Ag° metallic species, respectively. The in vitro viability of HaCaT cells after 24 h interaction with glass samples indicated good proliferation rates, excluding any cytotoxicity effect.

Microbiological examination of the skin’s bacterial flora from the BG-Vaseline, BGAuSP-Vaseline and BGAuIND-Vaseline treated batches taken from the area of choice of surgery prior to this procedure did not reveal pathological changes in the bacterial flora. Also, no pathological changes of the bacterial flora were found in the samples taken on the 7th post-intervention day, so can be concluded that there were no microbiological factors that negatively influence the healing of the wounds and that the ointments tested did not influence the bacterial flora.

Examination of the immune status of rats before surgery did not show pathological changes in their immunity, which may influence wound healing; also, in the assessment of the immune status of rats performed on the 7th post-intervention day no changes were found, concluding that there were no immune factors influencing wound healing and that the surgery itself with all procedures did not change in the negative, the immunity of rats.

Histopathological examination of the skin on day 8 postoperatively indicates faster healing in the BGAuSP-Vaseline-treated group than in the BG-Vaseline and BGAuIND-Vaseline-treated groups.

In conclusion, histopathological examination of the skin defects treated with the experimental product unequivocally demonstrates that the use of ointments with bioactive glass content enriched with spherical gold nanoparticles (BGAuSP-Vaseline) and spherical gold nanocages (BGAuIND-Vaseline) has a beneficial effect on the process of skin healing and regeneration, the created skin defects being completely healed compared to those in the control lots. However, the difference appears in the regeneration speed, the silver content of the BGAuIND slows the wound healing in the first week.

## Figures and Tables

**Figure 1 molecules-26-00620-f001:**
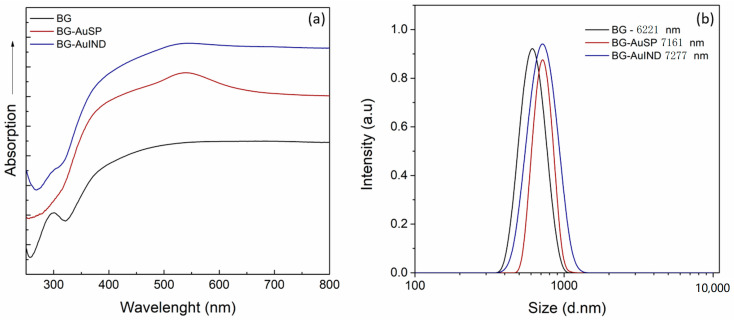
UV-Vis absorption spectra of the BG, BG-AuSP and BG-AuIND composites (**a**) and particle size distributions of BG, BG-AuSP and BG-AuIND measured by DLS (dynamic light scattering) (**b**).

**Figure 2 molecules-26-00620-f002:**
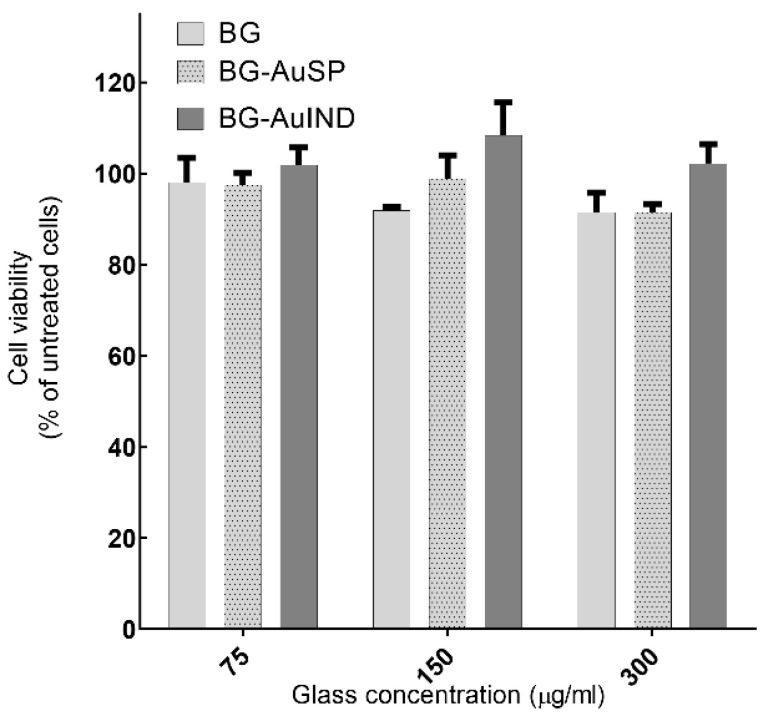
Viability of HaCaT cells after 24 h interaction with different concentrations of BG, BG-AuSP and BG-AuIND (*p* < 0.05).

**Figure 3 molecules-26-00620-f003:**
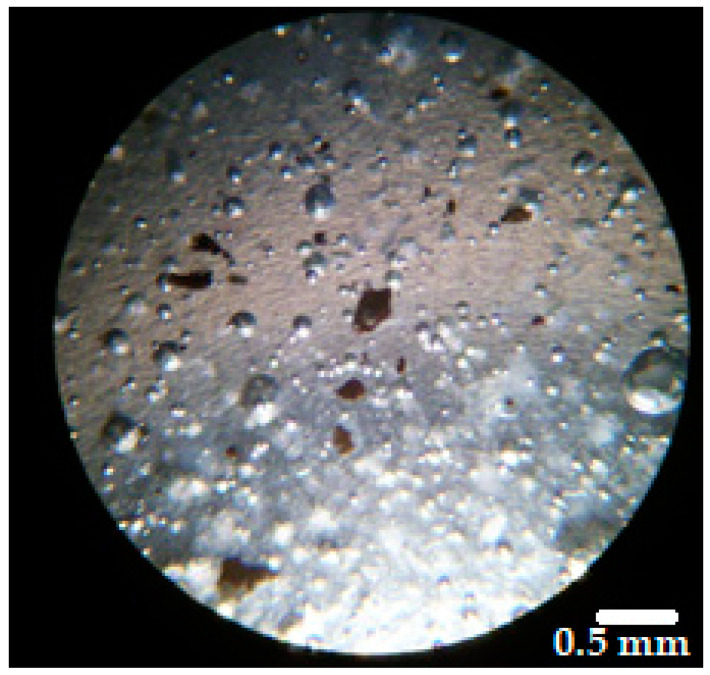
Microscopical image of the distribution of BGAuSP in the Vaseline ointment (magnification 40×).

**Figure 4 molecules-26-00620-f004:**
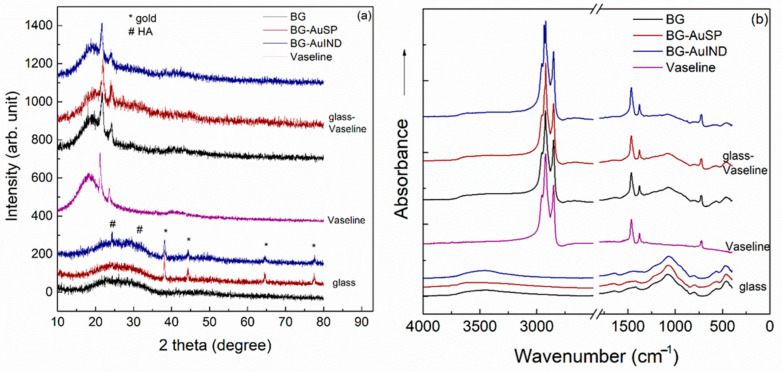
X-Ray Diffraction (XRD) patterns (**a**) and Fourier transform infrared (FTIR) spectra (**b**) of the BG and BG-AuSP, BG-AuIND, Vaseline, BG-Vaseline, BG-AuSP-Vaseline and BG-AuIND-Vaseline composites.

**Figure 5 molecules-26-00620-f005:**
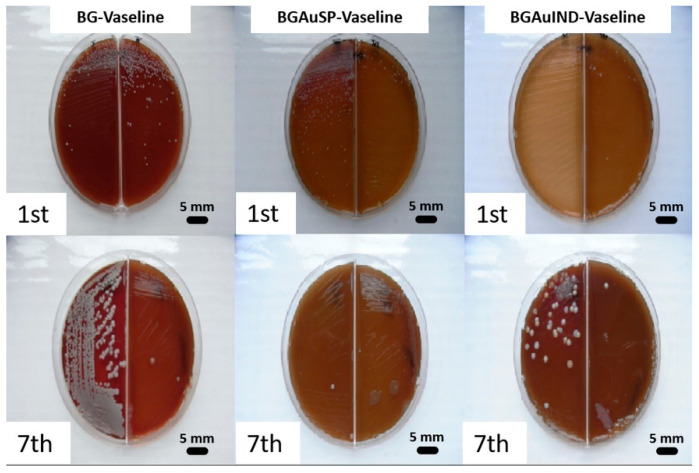
Representative images of bacterial cultures collected from the rats treated with BG-Vaseline, BGAuSP-Vaseline and BG-AuIND-Vaseline ointments on the 1st day of intervention and after 7th days post-surgery.

**Figure 6 molecules-26-00620-f006:**
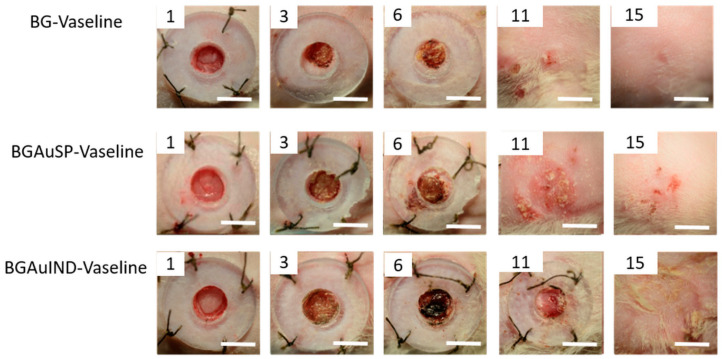
Representative images of full-thickness skin defect in rats with BG-Vaseline, BGAuSP-Vaseline and BG-AuIND-Vaseline ointments at 1, 3, 6, 11 and 15 days post-surgery. (Scale bar 6 mm).

**Figure 7 molecules-26-00620-f007:**
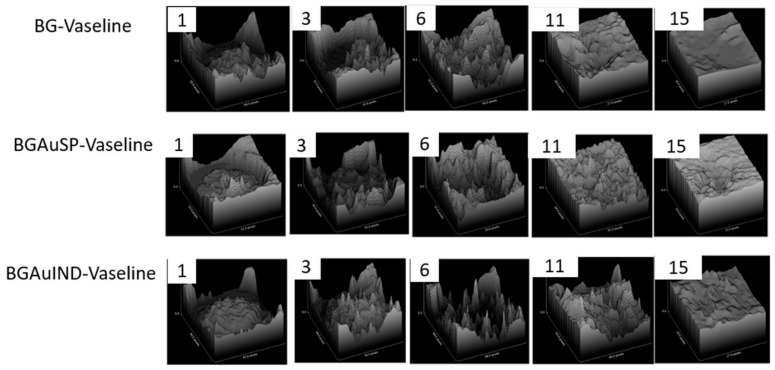
The evolution of wound healing presented in 3D at 1, 3, 6, 11, 15 post-surgery. (Scale bare 6 mm).

**Figure 8 molecules-26-00620-f008:**
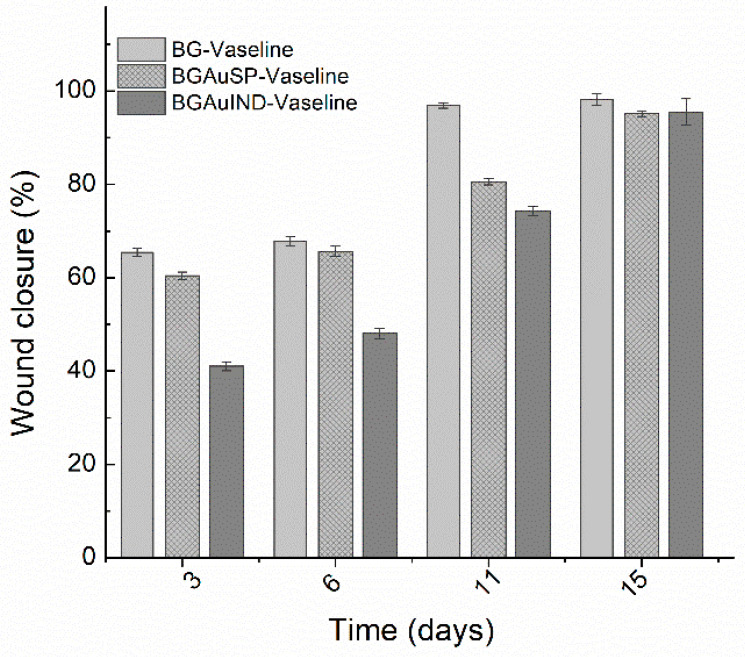
Percent of wound closure for the defect treated with BG-Vaseline, BGAuSP-Vaseline and BG-AuIND-Vaseline ointments. *p* < 0.05.

**Figure 9 molecules-26-00620-f009:**
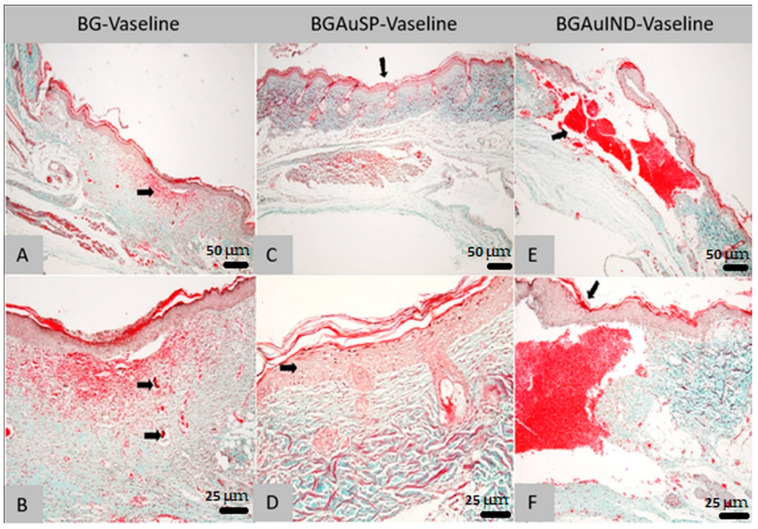
Histological findings of skin: the aspect of the skin in the area of the cutaneous defect; dermal granulation tissue with associated bleeding (arrow; Goldner trichrome col., ob. 4×) from the animals treated with BG-Vaseline (**A**,**B**), BGAuSP-Vaseline (**C**,**D**) and BGAuIND (**E**,**F**). (**A**)—Skin appearance in the area of skin defect; dermal granulation tissue with associated hemorrhage (arrow; trichrome Goldner-magnification, ob. 10×). (**B**)—Skin: completely regenerated epidermis, with the presence of some subepidermal spaces; In the dermis, granulation and hemorrhage tissue are identified, respectively the presence of an acidophilic-amorphous material without inflammatory reaction at the interface area with it (arrows; trichrome Goldner-magnification, ob. 10×). (**C**)—Skin: the aspect of the skin in the area of the cutaneous defect (arrow; Goldner trichrome col., ob. 10×). (**D**)—Skin: slight acanthosis (arrow) epidermis on the recuperative background and the presence of relatively well-differentiated connective tissue (Goldner trichrome col., ob. 40×). (**E**)—Skin: general aspect in the area of the defect (arrow), with sub epidermal hemorrhage (col. Goldner trichrome; ob. 4×). (**F**)—Skin: regenerated epidermis covering the defect area (arrow); discrete leukocyte infiltrate in the marginal areas of the defect (Goldner trichrome col., ob. 20×).

**Figure 10 molecules-26-00620-f010:**
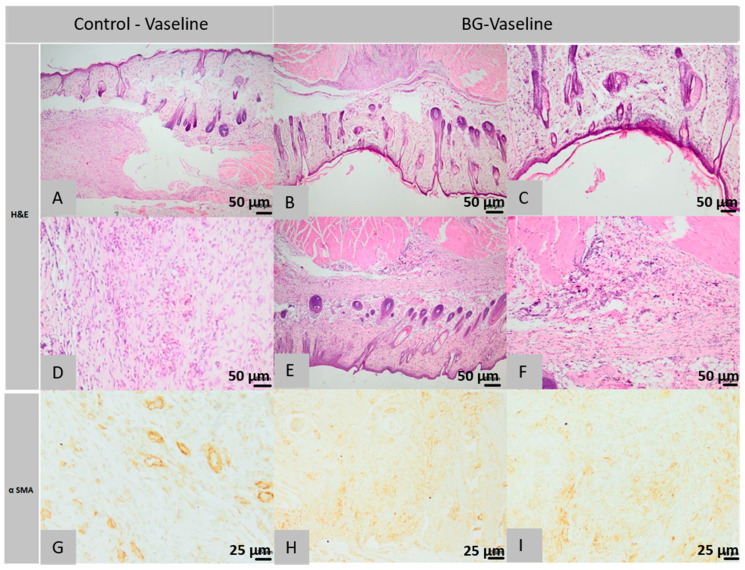
Histopathological images of the skin and subcutaneous tissue from the control animals of the group BG-Vaseline: granulation tissue with few histiocytes (**A**), blood vessels and myofibrils (**A**), abundant granulation tissue, partially oriented (**D**), histiocytes centered around the hair follicle (**D**) (col. HE, ob. 40×). Histopathological images of the skin and subcutaneous tissue from the animals treated with BG-Vaseline: the normal epidermis and collagen fibers in dermis (**B**,**C**), granulation tissue with blood vessels and myofibers (**E**), the surgical wound completely healed, presence of skin muscle (**F**) (col. HE, ob. 10×), at level superficial dermis fibroblasts, lymphocytes, histiocytes and mast cells, granulation tissue described above contains numerous blood vessels with fusiform cells (myofibroblasts) (**G**), (**H**), scar tissue with blood vessels and fusiform cells (**I**) (col. αSMA, ob. 10×).

**Figure 11 molecules-26-00620-f011:**
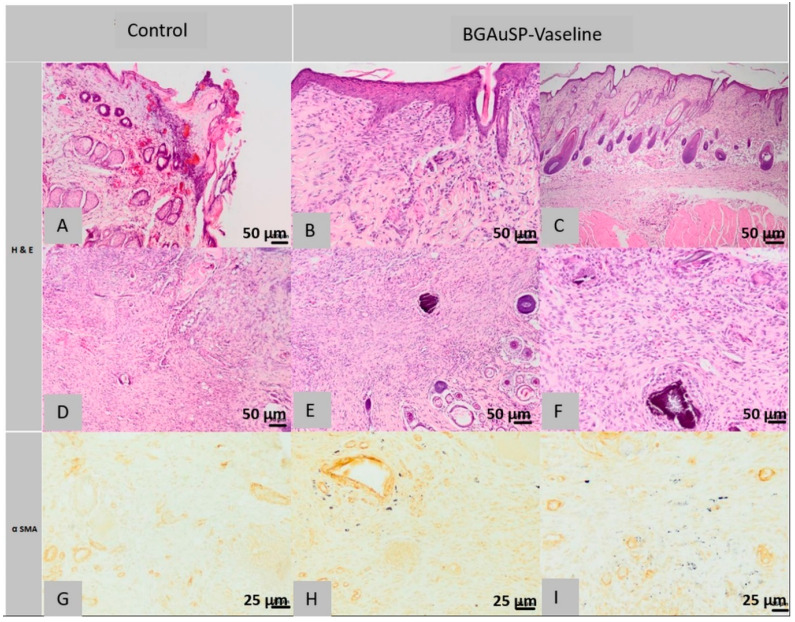
Histopathological images of the skin and subcutaneous tissue from the control animals from group BGAuSP-Vaseline: partially healed surgical wound (**A**), discontinuous epidermis in the process of regeneration (**A**), scar tissue contains few positive cells (**D**), fully cured surgical wound, partially regenerated pilosebaceous glands (**D**) (col. HE, ob. 40×), granulation formed by fusiform cells (**G**) (col. α SMA ob. 40×). Histopathological images of the skin and subcutaneous tissue from the animals treated with BGAuSP-Vaseline: the surgical wound was completely healed with excess granulation tissue containing multifocal granules of inert foreign body (**B**), granulation tissue formed by fusiform cells (**C**), in the deep dermis, numerous eosinophils are present with rare mast cells (**E**), the surgical wound completely healed with fibroblasts (**F**), (col. HE, ob. 40×) near the defect is interrupted and replaced with scar tissue infiltrated with rare mast cells (**H**) (col. Α SMA, ob. 40×), in the granulation tissue that replaces the dermis an *Panniculus carnosus* are fusiform (myofibroblast) positive α-SMA cells (**I**).

**Figure 12 molecules-26-00620-f012:**
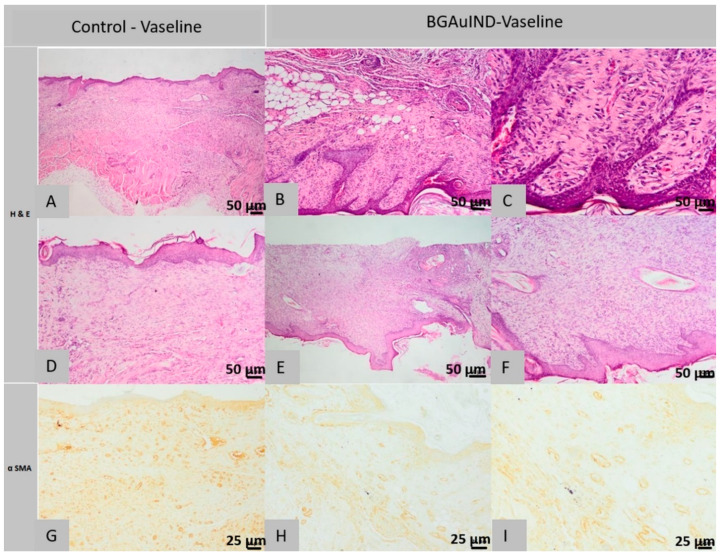
Histopathological images of the skin and subcutaneous tissue from the control animals from group BGAuIND-Vaseline: completely healed surgical wound, absent pilosebaceous glands (**A**), excess granulation tissue with fusiform cells, most myofibroblasts (**A**) and (**D**), (HE column, ob. 40×) epidermis, focal, discontinuous (**G**), excess granulation with myofibroblasts (**G**) (col. α SMA ob. 40×) Histopathological images of the skin and subcutaneous tissue from the animals treated with BGAuIND-Vaseline: completely healed surgical wound, partially regenerated pilosebaceous units (**B**,**C**), scar tissue with rare myofibers (**E**), completely healed surgical wound (**F**), (HE column, 40× ob.) excess granulation tissue, partially oriented (**H**), scar tissue with rare myofibrils (**I**) (col. α SMA ob. 40× This section may be divided by subheadings. It should provide a concise and precise description of the experimental results, their interpretation as well as the experimental conclusions that can be drawn.

**Figure 13 molecules-26-00620-f013:**
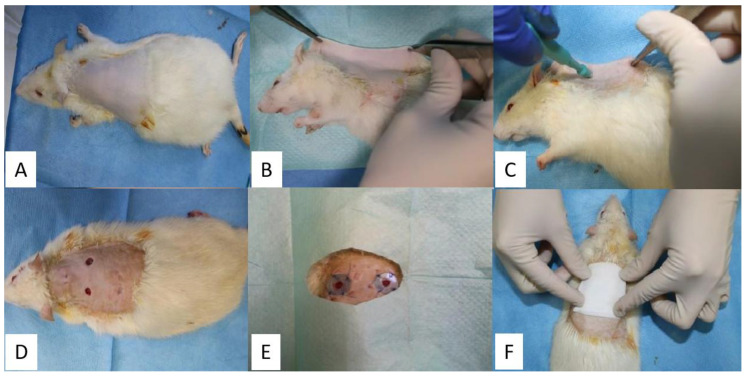
Obtaining the two cutaneous excisions and application of silicate rings and the ointment: antiseptic area (**A**), lateral positioning (**B**), using the biopsy-punch for excision formation (**C**), opening the fold to obtain two excisions (**D**), suture of silicate rings (**E**) application of a sterile pad (**F**).

**Table 1 molecules-26-00620-t001:** In vivo-post-therapeutic blastic transformation test.

Samples	Control (SI%)	PHA M (SI%)	ConA (SI%)	LPS (SI%)
BG-Vaseline	81.13 ± 0.53	81.24 ± 0.42	81.76 ± 0.73	81.66 ± 0.42
BGAuSP-Vaseline	74.8 ± 2.64	73.32 ± 0.1	72.59 ± 2.53	74.17 ± 1.16
BGAuIND-Vaseline	81.13 ± 0.53	81.76 ± 0.31	80.5 ± 1.58	82.08 ± 0.42

## Data Availability

Data is contained within the article.

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
