# Peer review of "The Impact of Composites with Silicate-Based Glasses and Gold Nanoparticles on Skin Wound Regeneration"

_molecules, 2021, doi:10.3390/molecules26030620_

Round 1

Reviewer 1 Report

However, it is well written article with a lot of results, I have some comments/recommendation:

  • In line 67 is written: “ Silver nanoparticles are used for the synthesis of the spherical nanocages (AuIND)……”. Why the authors did not use also the silver nanoparticles for the viability evaluation? And I also miss the characterization of silver nanoparticles.
  • In the section “Materials and Methods” I suggest to change the paragraph 3.2.2. (in vitro cell viability..) and 3.2.3 (Glass composition..).
  • In paragraph 3.2.3. “Glass composition ointments formation” is not described how was prepared AuIND with silver.

Author Response

However, it is well written article with a lot of results, I have some comments/recommendation:

  • In line 67 is written: “ Silver nanoparticles are used for the synthesis of the spherical nanocages (AuIND)……”. Why the authors did not use also the silver nanoparticles for the viability evaluation? And I also miss the characterization of silver nanoparticles.

A: In our previous study was demonstrated that spherical gold nanocages (AuIND) promote the proliferation rate of human keratinocyte cells [Magyari, K.; Tóth, Z.R.; Pap, Z.; Licarete, E.; Vodnar, D.C.; Todea, M.; Gyulavári, T.; Hernadi, K.; Baia, L. Insights into the effect of gold nanospheres, nanotriangles and spherical nanocages on the structural, morphological and biological properties of bioactive glasses. J. Non. Cryst. Solids 2019, 522, 119552, doi:10.1016/j.jnoncrysol.2019.119552]. The silver nanoparticles were used only as a template of Au nanoparticle’s synthesis. The same synthesis approach was used in our previous articles for the synthesis of Ag nanoparticles, where we have obtained silver nanoparticles within 30-80 nm particle size (confirmed by TEM and DRS measurement) [Tóth, Z.R.; Kovács, G.; Hernádi, K.; Baia, L.; Pap, Z. The investigation of the photocatalytic efficiency of spherical gold nanocages/TiO2and silver nanospheres/TiO2composites. Sep. Purif. Technol. 2017, 183, 216–225, doi:10.1016/j.seppur.2017.03.065]. The obtained ratio was Ag:Au=1:99. Since the Ag nanoparticles in 99% were replaced with gold nanoparticles by the replacement synthesis method and based on the large manuscript draft, we have not focused on Ag nanoparticles characterization.

  • In the section “Materials and Methods” I suggest to change the paragraph 3.2.2. (in vitro cell viability..) and 3.2.3 (Glass composition..).

A: The paragraph 3.2.2 and 3.2.3 has been changed according to reviewer suggestion. 

  • In paragraph 3.2.3. “Glass composition ointments formation” is not described how was prepared AuIND with silver.

A: The synthesis of gold nanocages (AuIND) has been described in section 3.2.1 The synthesis and characterization of glass composites with AuNPs.

“The spherical gold nanocages (AuIND) were prepared by adapting a well-known method in the literature [Tóth, Z.R.; Kovács, G.; Hernádi, K.; Baia, L.; Pap, Z. The investigation of the photocatalytic efficiency of spherical gold nanocages/TiO2and silver nanospheres/TiO2composites. Sep. Purif. Technol. 2017, 183, 216–225]. The Ag nanoparticles were prepared via chemical reduction method, where NaBH4 was used as a reducing agent and trisodium citrate as a stabilizing agent. The next step was the galvanic replacement reaction between the as-obtained Ag nanoparticles and HAuCl4·3H2O solution. NaCl was used to eliminate the remaining Ag+ in form of AgCl.”

Reviewer 2 Report

The manuscript deals with the impact of silica and gold nanoparticles incorporated to Vaseline ointments for skin wound regeneration. It is well written and structured. After corrections of some minor issues, I recommend publishing the manuscript.

Comments, questions and notes are the following:

Results and discussion: the authors are after the 2.4 section just describing results without any discussion. Could the authors provide also discussion?

How the silver content in the BGAuIND samples slows down the wound healing in the first week?

Some formal mistakes:

Line 26 -  but the in the meantime (removing the)

Line 33 – missing dot

Line 71 – 18 % (weight or volume)?

Figures 3, 5, 7 and 9 – missing the scale bar.

Section 3.7 – wrong formatting.

Author Response

The manuscript deals with the impact of silica and gold nanoparticles incorporated to Vaseline ointments for skin wound regeneration. It is well written and structured. After corrections of some minor issues, I recommend publishing the manuscript.

Comments, questions and notes are the following:

Results and discussion: the authors are after the 2.4 section just describing results without any discussion. Could the authors provide also discussion?

A: The sections 2.4 and 2.5 were completed with followed discussions:

In section 2.4:

“Comparing the results obtained from histopathological examination of the skin on day 8 postoperatively shows that the wound of the group treated with BGAuSP-Vaseline heals faster than the BG-Vaseline and BGAuIND-Vaseline-treated groups.’’

In subsection 2.5.1:

‘’In the group treated with BG-Vaseline ointment, the wound is completely healed. The epidermis is normal. The dermis has focal infiltrates with rare lymphocytes, plasma cells, and mast cells. The pilosebaceous units are partially regenerated. In the control group, however the surgical wound is not completely healed. The dermis and muscle panniculus carnosus is replaced by abundant granular tissue, partially oriented.”

In subsection 2.5.2:

“In the group treated with BGAuSP-Vaseline ointment, the wound is completely healed. The epidermis has a normal structure. The dermis has numerous fibroblasts oriented parallel to the dermis, and the pilosebaceous units are completely regenerated. While the wound is partially healed in the control group, the epidermis is focally discontinuous, the dermis is replaced by abundant scar tissue, and the pilosebaceous units are partially regenerated”

In subsection 2.5.3:

‘’In the group treated with BGAuIND-Vaseline ointment, the epithelium is moderately hyperplasia. The pilosebaceous units are partially regenerated, and the collagen fibers are oriented parallel to the epidermis. While in the control group, the pilosebaceous units are absent. The dermis and the panniculus carnosus muscle are replaced with excess granulation tissue, partially oriented.’’

How the silver content in the BGAuIND samples slows down the wound healing in the first week?

A: Although the strong antibacterial effect of AgNP is well known, excessive agglomeration of these nanoparticles sometimes leads to cell death [Ji Hong Min, Madhumita Patel, Won-Gun Koh, Incorporation of Conductive Materials into Hydrogels for Tissue Engineering Applications, Polymers 2018, 10, 1078; doi:10.3390/polym10101078]. The manuscript was completed with this information.

Some formal mistakes:

Line 26 - but the in the meantime (removing the)

A: The “the” was removed.

Line 33 – missing dot

A: The dot was inserted at the end of sentence.

Line 71 – 18 % (weight or volume)?

A: The BG concentration in the Vaseline is 18 wt%. (see line 68)

Figures 3, 5, 7 and 9 – missing the scale bar.

A: The scale bar was added for all indicated figures.

Section 3.7 – wrong formatting.

A: The format was carefully checked and is fine.

Reviewer 3 Report

The paper is interesting, the figures are very informative and of satisfactory quality. However, something that I believe to be a pretty good science is buried under poor language style

The language has to be improved prior to publication. I would advise avoiding very long and complex sentences, e.g. line 27-30 or 698-703 to list a few. 

Author Response

The paper is interesting, the figures are very informative and of satisfactory quality. However, something that I believe to be a pretty good science is buried under poor language style.

The language has to be improved prior to publication. I would advise avoiding very long and complex sentences, e.g. line 27-30 or 698-703 to list a few.

A: In accordance with requirements of the reviewer the language was improved.